# UNCERTAINTY-AWARE REWARD MODEL: TEACHING REWARD MODELS TO KNOW WHAT IS UNKNOWN

## ABSTRACT

Reward models (RM) play a critical role in aligning generations of large language models (LLM) to human expectations. However, prevailing RMs fail to capture the stochasticity within human preferences and cannot effectively evaluate the reliability of reward predictions. To address these issues, we propose Uncertain-aware RM (URM) and Uncertain-aware RM Ensemble (URME) to incorporate and manage uncertainty in reward modeling. URM can model the distribution of disentangled attributes within human preferences, while URME quantifies uncertainty through discrepancies in the ensemble, thereby identifying potential lack of knowledge during reward evaluation. Experiment results indicate that the proposed URM achieves state-of-the-art performance compared to models with the same size, demonstrating the effectiveness of modeling uncertainty within human preferences. Furthermore, empirical results show that through uncertainty quantification, URM and URME can identify unreliable predictions to improve the quality of reward evaluations.

## 1 INTRODUCTION

Large language models (LLM) have demonstrated remarkable capabilities across various domains (Singhal et al., 2023a; Cui et al., 2024; Kasneci et al., 2023). These powerful LLMs are trained to align with human values and expectations to avoid harmful and toxic generations. To achieve alignment, LLMs rely on feedbacks from reward models (RM), where the feedbacks are provided in the form of rewards (Singhal et al., 2023a; Cui et al., 2024; Kasneci et al., 2023). These rewards typically reflect the quality and users' preferences of the responses provided, and hence reward maximization will guide the LLM to more effectively satisfy user queries. In this paradigm, RMs fundamentally decides the efficacy of alignment, as they primarily steer the LLMs through feedback. Therefore, the reliability and accuracy of this feedback is essential in aligning LLMs with intended human values and preferences.

However, current RMs fail to capture the stochastic nature of human preferences (Baylis, 1950) and lack the ability to evaluate the reliability of the predicted rewards. In prevalent RMs, a value head (usually a linear layer) is added to the pretrained base model and maps the hidden states to reward scalars (Bai et al., 2022a; Ouyang et al., 2022) or attribute scores (Adler et al., 2024; Wang et al., 2024a). This results in a deterministic reward modeling process, unable to accommodate the variabilities of human preferences. Moreover, there is no other information to validate the reliability of these reward predictions.

Uncertainty is of major importance in machine learning (Hüllermeier & Waegeman, 2021) and an appropriate representation for uncertainty is essential for developing trustworthy and reliable models (Yang et al., 2009; Varshney & Alemzadeh, 2017). Uncertainty originates from two different sources: *aleatoric* and *epistemic*. Aleatoric uncertainty refers to the inherent variability and randomness of data. As opposed to this, epistemic uncertainty is caused by lack of knowledge, i.e. ignorance of the model instead of any underlying randomness.

In the context of reward modeling for LLM, aleatoric uncertainty refers to the stochasticity of human preferences, while epistemic uncertainty comes from the RMs' lack of knowledge to make reliable evaluations. Therefore, introducing uncertainty to reward modeling improves modeling capacity of RMs and enhance reliability of the reward predictions. Consequently, by identifying and filtering

out out-of-distribution (OOD) data where RMs fail to generalize, rewards with better reliability pave the way for more efficient alignments of LLMs.

In this paper, we propose Uncertain-aware RM (URM) and Uncertain-aware RM Ensemble (URME) to handle aleatoric and epistemic uncertainty in reward modeling respectively. URM is equipped with an uncertainty-aware value head to model the distributions of multiple attributes within human preferences. We demonstrate that with the popular bradley-terry-model loss function (Bradley & Terry, 1952), RMs cannot quantify uncertainty of human preferences even with an uncertainty-aware value head. Therefore, URMs are trained via maximum likelihood estimation and attribute regression. URME quantifies epistemic uncertainty by the discrepancies among URMs in the ensemble, identifying potential lack of knowledge. During reward evaluation, filtering strategy can be applied to prompt-response pairs with high uncertainty, in case that LLMs learn unintended or potentially harmful behaviors that URMs may not be able to accurately evaluate.

Empirical results on a popular RM benchmark RewardBench (Lambert et al., 2024) demonstrate that URM with 8B model size achieves state-of-the-art performance among models with the same size and outperforms a number of strong large models including Nemotron-4-340B (Adler et al., 2024). And through uncertainty quantification, URM and URME are able to identify their level of knowledge for the input data and make the reward evaluations more reliable through filtering strategy. Furthermore, results of best-of-$n$ sampling validates that URM and URME can effectively enhance the generation quality of LLMs.

Contributions of this paper include:

(1) We introduce URM and URME to model the uncertainty within human preferences and reward models themselves.

(2) URM and URME are able to improve LLMs' generation effectively. Notably, URM achieves state-of-the-art performance on RewardBench compared with models of the same size (8B).

(3) Empirical results demonstrate that URM and URME can successfully quantify uncertainty to identify areas where the models lack sufficient knowledge to make accurate predictions, leading to more reliable reward evaluations.

## 2 PRELIMINARIES

LLM alignment typically consists of three stages (Ouyang et al., 2022): supervised fine-tuning (SFT), reward modeling and proximal policy optimization (PPO) (Schulman et al., 2017). SFT utilizes expert demonstrations to fine-tune the pretrained base model in a supervised-learning fashion to enable LLMs to follow user instructions.

**Reward Modeling** Reward modeling aims to learns human preferences explicitly (Ouyang et al., 2022) or implicitly (Rafailov et al., 2024). For some prompt $x$ and a response pair $(y_w, y_l)$, $y_w$ is the chosen response preferred by humans and $y_l$ is rejected. Following the Bradley-Terry model (Bradley & Terry, 1952), under RM $r_\phi$ the probability of $y_w$ being preferred than $y_l$, i.e. $y_w \succ y_l$, is

$$P(y_w \succ y_l | x) = \log \frac{\exp(r_\phi(x, y_w))}{\exp(r_\phi(x, y_w)) + \exp(r_\phi(x, y_l))} \tag{1}$$
$$= sigmoid(r_\phi(x, y_w) - r_\phi(x, y_l))$$

Thus, to train a RM to prioritize chosen responses over rejected responses, the loss function is the maximum likelihood estimation of Eq. 1

$$L_1 = -\mathbb{E}_{x, y_w, y_l \sim D} \left[ \log sigmoid \left( r_\phi(x, y_w) - r_\phi(x, y_l) \right) \right], \tag{2}$$

where $r_\phi$ is the reward model parameterized by $\phi$, consisting of the pretrained base model and a linear value head. The trained RM can be used to improve the LLM's generation by Best-of N (BoN) (Stiennon et al., 2020) or RLHF (Ouyang et al., 2022).

**PPO** In this stage, LLMs are fine-tuned with feedbacks from the RM. To prevent the model deviate too far from the pretrained model and forget linguistic skills, there is also an Kullback-Leibler (KL) divergence penalty in the reward from the RM. Thus, the total reward $\hat{r}$ is

$$\hat{r}(x, y) = r_\phi(x, y) - \eta \text{KL}(\pi(y|x) \| \pi_{\text{ref}}(y|x)), \tag{3}$$

where $\eta$ is the coefficient for the KL penalty, $\pi$ is the model to be fine-tuned and $\pi_{\text{ref}}$ is the reference model which is usually the SFT model. Running PPO to maximize reward from Eq. 3 can not only align the LLM with human preferences, but also prevent it from severely deviating from the reference model.

# 3 RELATED WORK

## 3.1 MULTI-ATTRIBUTE REWARD MODELING

To generate helpful, harmless and truthful responses (Askell et al., 2021), LLMs must be aligned with human expectations. Current methods fine-tune models based on human (Christiano et al., 2017; Stiennon et al., 2020; Bai et al., 2022a; Ouyang et al., 2022) or AI feedbacks (Bai et al., 2022b; Sun et al., 2024) to maximize preference-based rewards, which are provided by reward modeling. Typically, a reward model is learned and LLMs will improve their generation quality according to feedbacks from the reward model (Bai et al., 2022a; Ouyang et al., 2022; Shao et al., 2024; Stiennon et al., 2020).

Recent studies show that human and LLM judges may introduce potential biases to annotations of preference (Zhang et al., 2023; Kotek et al., 2023; Wang et al., 2024b; Chen et al., 2024a). Moreover, traditional RMs usually rely on single-dimensional feedback on general quality instead of fine-grained multifaceted signals to indicate multiple attributes such as helpfulness, coherence and verbosity (Dong et al., 2023b). Adler et al. (2024) discovered that multi-attribute RMs trained on datasets with high-quality attribute-specific annotations (Cui et al., 2023; Wang et al., 2024c) are able to disentangle real helpfulness and other irrelevant aspects such as lengthy bias (Shen et al., 2023; Singhal et al., 2023b). There are also alignment methods directly aimed at multi-attribute alignment. Zhou et al. (2023) includes preference on multiple attributes in the Direct Preference Optimization (DPO) loss function (Rafailov et al., 2024), trying to optimize preference rewards for all attributes simultaneously. Lou et al. (2024) proposed to achieve multi-attribute alignment sequentially, one attribute at a time, where LLMs learns to align with new attributes while staying aligned with previous dimensions.

## 3.2 RLHF, OFFLINE RL AND UNCERTAINTY

In RLHF, LLM policy is optimized via interactions with the RM, whose training data is pre-collected preference pairs (Bai et al., 2022a; Ouyang et al., 2022). In this setting, RLHF falls into the category of offline RL, where RL policies cannot interact with the environment and get feedbacks in real time, but instead can only be updated based on an offline dataset collected by some other policy (Levine et al., 2020). Offline RL is notoriously difficult due to the distributional shift issue (Lou et al., 2022; Ma et al., 2021; Prudencio et al., 2023). Recent advancements in iterative LLM alignment methods (Yuan et al., 2024; Dong et al., 2024; Xiong et al., 2024) iterates between LLM fine-tuning and the sampling and annotation of new training data, alleviating the distributional shift issue. Although these iterative methods aim to transcend the constraints of the offline setting,, RLHF is still offline within each iteration.

An important topic in offline RL is uncertainty quantification, which has long underpinned many critical roles (Abdar et al., 2021), such as trustworthy decision-making (Huang et al., 2019; Eriksson & Dimitrakakis, 2019) and improving reliability of machine learning models (Wang et al., 2019). In offline RL, uncertainty quantification enables out-of-distribution data detection and keep the policy within the offline dataset's support area through conservative updates to avoid distributional shift (Yu et al., 2020; Kidambi et al., 2020; An et al., 2021; Zhu et al., 2024). So it is natural to introduce uncertainty to RLHF to make LLM alignment more reliable and effective.

Ensemble of RMs are discussed in previous works. However, we study the ensemble of uncertainty-aware RMs to identify unreliable reward evaluations, while previous discussions are limited to using RM ensembles to mitigate reward hacking Coste et al. (2023); Eisenstein et al. (2023) and using value heads to disentangle length and quality in reward modeling Chen et al. (2024b). We notice a concurrent work QRM Dorka (2024) which also models human preferences by distributions. QRM only studies distributional RMs, while we also study the ensemble of such uncertainty-aware RMs. Moreover, QRM is trained via quantile regression (Koenker, 2017), a variant of our attribute regres-

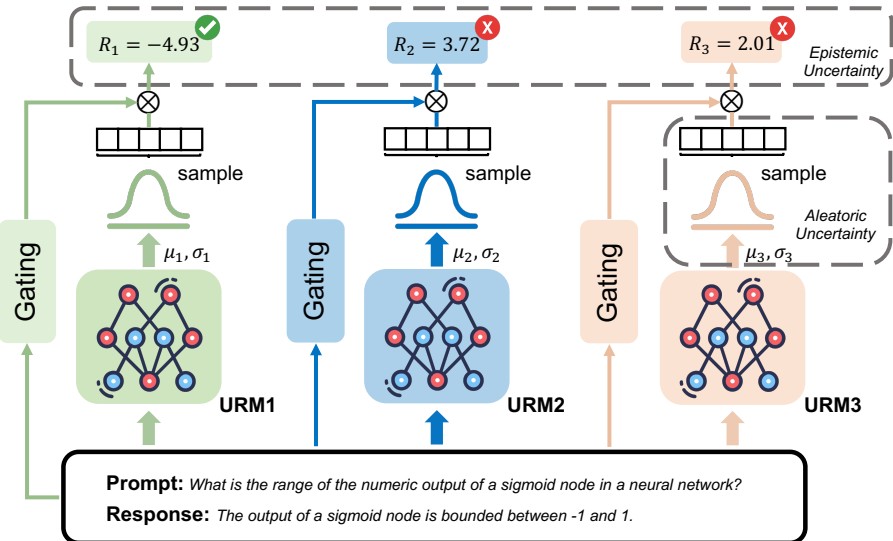

Figure 1: Architecture of URM and URME. URMs output $\mu$ and $\sigma$ to parameterize normal distributions, from which multi-attribute scores are sampled. The scores are then combined to reward scalars by weights generated by a gating layer. URME consists of multiple URMs, allowing for quantification of the epistemic uncertainty using the disagreement among the URMs. In the given example, there is a substantial divergence among URMs, indicating significant epistemic uncertainty. Thus, although URM 1 correctly assigns a small, negative reward to the input, the significant epistemic uncertainty still indicates the URMs lack relevant knowledge to provide reliable evaluation of the inputs.

sion. But we also studied uncertainty-aware RMs trained via maximum likelihood estimation, which can better capture the uncertainty of rewards.

## 4 METHODOLOGY

In this section, we will introduce our uncertain-aware reward model (URM) and uncertainty-aware reward model ensemble (URME) to quantify aleatoric and epistemic uncertainties respectively.

Fig. 1 gives the architecture of URM and URME. URMs quantify the aleatoric uncertainty by modeling the distribution of scores, and the epistemic uncertainty is quantified by the disagreement within the URME. In the given example, the response is incorrect and there is large disagreement within the URME, indicating significant epistemic uncertainty and the models' lack of relevant knowledge.

### 4.1 UNCERTAINTY-AWARE REWARD MODEL

Traditional RMs optimize the Bradley-Terry model loss (BT-loss) in Eq. 2 to enlarge the discrepancy between the rewards of chosen and rejected responses so that the preference probability is maximized. The value head will map hidden states from the base model to a scalar reward. Such mapping is deterministic and thus cannot catch any uncertainty (Chua et al., 2018) within the reward modeling process.

However, at its core, human preferences exhibit a distinctly probabilistic nature, rather than being strictly deterministic (Baylis, 1950). This issue is further exaggerated due to the bias and inconsistencies introduced by human annotators (Sylolypavan et al., 2023; Sleeman & Gilhooly, 2023; Chen et al., 2024a). Between individuals, preferences differ from person to person. This means what's preferable for one may not be for another. Even within individuals, preferences are not static. They can swing based on numerous factors such as mood and context. These stochastic natures of

human preferences contribute to adopting a probabilistic framework for modeling preferences with aleatoric uncertainty.

Prior works have explored a number of uncertainty-aware neural networks (Neal, 2012; Lakshminarayanan et al., 2017), especially in RL (Gal et al., 2016; Depeweg et al., 2016) and model-based RL (MBRL) (Chua et al., 2018; Yu et al., 2020; Kidambi et al., 2020).

Considering RMs act similarly to the reward part of the dynamics model in MBRL, aleatoric uncertainty within human preferences can be captured by outputting the parameters of a parameterized distribution. Specifically, unlike traditional RMs that output a single deterministic reward value, uncertainty-aware RMs can model the distributions of human preferences. As schematically shown in Fig. 2, given a prompt-response pair with multiple preference annotation samples, traditional RMs can only provide a fixed reward estimation and fails to represent the real preference. But uncertainty-aware RMs are able to offer a more accurate approximation of the human preference distribution.

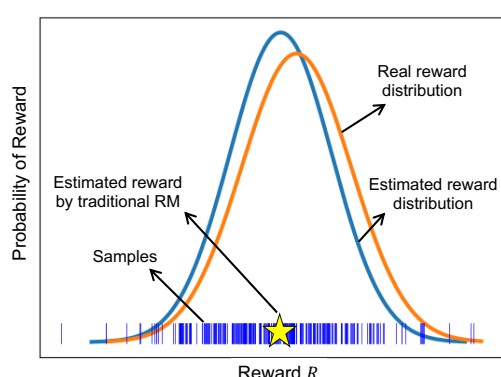

Figure 2: Comparison between URM and traditional RM in estimating preference distribution.

To model the preference reward distribution, URM adds an probabilistic value head to the pretrained base model. The value head takes in the last hidden state $h$ of the base model and outputs mean $\mu$ and logged standard deviation $\sigma$ to parameterize a normal distribution $\mathcal{N}(\mu, \exp(2\sigma))$, from which preference rewards are sampled, and the aleatoric uncertainty is quantified by variance of the distribution. Reparameterization technique is adopted to enable gradient back-propagation.

However, we show that introducing the probabilistic value head and the sample-based reward to RMs with BT-loss, the aleatoric uncertainty still cannot be quantified.

We denote the reparameterization parameter $\alpha \sim \mathcal{N}(0, 1)$, and thus the sampled reward $r = \mu + \alpha\exp(\sigma)$. Substituting chosen reward $r_w$ and rejected reward $r_l$ into BT-loss in Eq. 2. For some given input $x, y_w, y_l$, gradient w.r.t. the logged standard deviation $\sigma_w$ is given by

$$\nabla_{\sigma_w} L_1 = -\mathbb{E}_{\alpha_w,\alpha_l \sim \mathcal{N}(0,1)} \left[ \alpha_w \big(1 - sigmoid(r_w - r_l)\big)\exp(\sigma_w) \right] = 0,$$

where $r_w = \mu_w + \alpha_w\exp(\sigma_w)$ and $r_l = \mu_l + \alpha_l\exp(\sigma_l)$. Similarly, $\nabla_{\sigma_l} L_1 = 0$. The unlearning effect of the variance term demonstrates that under the BT-loss, RMs still cannot quantify the uncertainty even equipped with a probabilistic value head.

Recent advances in multi-attribute RMs demonstrate they are capable of providing fine-grained rewards and disentangling real helpfulness and other irrelevant aspects such as lengthy bias (Adler et al., 2024; Chen et al., 2024a). The multi-attribute scores consist of human-or-AI-annotated ratings on multiple aspects such as helpfulness, coherence and verbosity. To learn a multi-attribute uncertainty-aware RM, we propose two ways to train the probabilistic value head.

**Maximum Likelihood Estimation** In URM, scores of all attributes are modeled by a distribution, we can train the probabilistic value head with maximum likelihood estimation (MLE). Since attributes are disentangled in multi-attribute RMs, it is fair to assume that they are independent, i.e. diagonal covariance for the parameterized normal distribution. Thus, the MLE loss function for URM is

$$L_2 = -\mathbb{E}_{x,y\sim D} \left[\log \mathbf{P}_\theta(R|x,y)\right] = -\mathbb{E}_{x,y\sim D} \left[\sum_{i=0}^{n} \log P_\theta(R_i|x,y)\right], \tag{4}$$

where $R_i$ is the $i$-th attribute score from the label and $\log P_\theta(R_i|x,y)$ is the log-probility of $R_i$ from the parameterized distribution $\mathcal{N}(\mu_i, \exp(2\sigma_i))$. Though MLE, the probabilistic value head is able to efficiently approximate the attribute scores' distribution, hence training URMs to fit the unique characteristics of the attribute scores.

**Attributes Regression with Reparameterization** we can also directly regress the sample-based rewards on multi-attribute scores $R \in \mathbb{R}^n$, similar as Adler et al. (2024) but with sampling and reparameterization. In this setting, URM's mean square error (MSE) loss function is

$$L_3 = \mathbb{E}_{x,y \sim D} \left[ \sum_{i=0}^{n} (r_i(x,y) - R_i)^2 \right] \tag{5}$$

where $i$ indicates $i$-th attribute, and $r_i \sim \mathcal{N}(\mu_i, \exp(2\sigma_i))$ is sampled from the distribution parameterized by the output of the probabilistic value head. To enable gradient back-propagation, we use the reparameterization technique, so that $r = \mu + \alpha \exp(\sigma)$, where reparameterization parameter $\alpha \sim \mathcal{N}(0,1)$. A more detailed analysis of this MSE loss is given in the appendix B.

With the trained probabilistic value head, during inference we can use mean $\mu_i$ for each attribute $i$ as the scores. We learn a gating layer to combine the multi-attribute scores to a reward scalar via weighted sum (Wang et al., 2024a). The gating layer is a fully-connected network, whose input is the last hidden states of the base LLM. And the learning objective of the gating layer is to prioritize chosen responses over rejected responses through the BT loss. For some prompt $x$, chosen response $y_w$ and rejected response $y_l$, the gating layer will output weights $\omega$ to combine the scores and thus the loss function is given by

$$L_4 = -\mathbb{E}_{x,y_w,y_l \sim D} \left[ \log sigmoid \left( \mu^T(x,y_w)\omega(x,y_w) - \mu^T(x,y_l)\omega(x,y_l) \right) \right], \tag{6}$$

Since the gating layer only offers weights to combine the scores, the base model and the probabilistic value head are kept frozen during training the gating layer. Besides the gating layer, the multi-attribute scores can also be combined by prior weights and still demonstrate competitive performance (Adler et al., 2024).

## 4.2 UNCERTAINTY-AWARE REWARD MODEL ENSEMBLE

Bootstrap ensemble of models is simple and effective for epistemic uncertainty quantification compared with other methods (Neal, 2012; Hernández-Lobato & Adams, 2015; Blundell et al., 2015).

Fig. 3 illustrates how URME works in quantifying uncertainty. The input space $X \times Y$ ($X$ for prompt and $Y$ for response) is split into the known and unknown area. In the known area, the training dataset can well support URMs to make reliable reward evaluations. However, in the unknown area, the situation differs. Given that each model utilizes different weight initialization and is optimized with distinct data mini-batches, the ensemble models are likely to diverge, potentially leading to varying and inconsistent evaluations. This inconsistency and divergence among models indicates the degree of uncertainty in

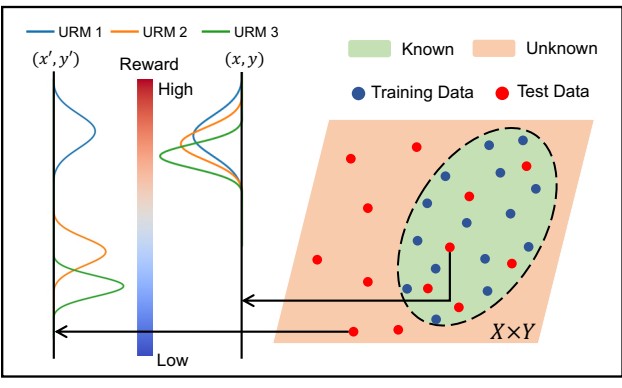

Figure 3: Illustration of URME in epistemic uncertainty quantification.

URME and large uncertainty in turn indicates the input data may belong to the unknown area.

Specifically, after obtaining distributions of the multi-attribute scores, the uncertainty can be measured by the largest discrepancy in URME

$$u_1(x,y) = \max_{i,j} \left( r^{(i)}(x,y) - r^{(j)}(x,y) \right), \tag{7}$$

where $i,j$ are URMs within the ensemble. Yu et al. (2020) proposed to capture both epistemic and aleatoric uncertainty by the largest variance in the ensemble

$$u_2(x,y) = \max_i (\|\Sigma^{(i)}(x,y)\|_F), \tag{8}$$

Table 1: Results on RewardBench. RewardBench evaluates four abilities: Chat, Chat-Hard (C-HARD), Safety and Reasoning. Rankings are decided by the overall average score of all categories.

| MODEL | BASE | SCORE | CHAT | C-HARD | SAFETY | REASON |
|---|---|---|---|---|---|---|
| URM (Ours) | Llama3.1-8B | 92.9 | 95.5 | 88.2 | 91.1 | 97.0 |
| SFR-Judge-r | Llama3.1-70B | 92.7 | 96.9 | 84.8 | 91.6 | 97.6 |
| Skywork-8B | Llama3.1-8B | 92.5 | 95.8 | 87.3 | 90.8 | 96.2 |
| Nemotron-RM | Nemotron4-340B | 92.0 | 95.8 | 87.1 | 91.5 | 93.6 |
| GRM | Llama3-8B | 91.5 | 95.5 | 86.2 | 90.8 | 93.6 |
| ArmoRM | Llama3-8B | 90.4 | 96.9 | 76.8 | 90.5 | 97.3 |
| InternLM2-RM | InternLM2-20B | 90.2 | 98.9 | 76.5 | 89.5 | 95.8 |
| SteerLM-RM | Llama3-70B | 88.8 | 91.3 | 80.3 | 92.8 | 90.6 |
| Gemini-1.5-pro | - | 88.2 | 92.3 | 80.6 | 87.9 | 92.0 |
| GPT-4o | - | 86.7 | 96.1 | 76.1 | 88.1 | 86.6 |
| GPT-4-turbo | - | 86.0 | 95.3 | 74.3 | 87.6 | 86.9 |

where $\Sigma^{(i)}$ is the covariance of $i$-th URM, which is diagonal in our case. This uncertainty estimator quantifies uncertainties from both sources and works effectively in offline MBRL setting.

Accurate reward evaluation is crucial in LLM alignment, as it fundamentally steers the learning process. Thus, we can adopt a filtering strategy to discard data with highly uncertain reward evaluations, since RMs may exhibit poor generalization and lack sufficient knowledge to provide reliable feedbacks for them. In this way, we can prevent LLMs from learning undesired behaviors, promoting a more controlled and trustworthy alignment process.

## 5 EXPERIMENT

### 5.1 EXPERIMENT SETTINGS

In our experiment, URM is based on Llama3.1 with 8 billion parameters. Before adding the probabilistic value head, we initialize URM's base model with weights from Liu & Zeng (2024). The gating layer consists of two fully-connected layers with hidden size 4096 activated by SELU (Klambauer et al., 2017). More information on URM training and implementation is given in the appendix A.1, C.1. URME have 3 URMs with different random seeds, probabilistic value head initialization and mini-batches of training data.

We utilize HelpSteer 2 (Wang et al., 2024c) as the training dataset to train the base model and the probabilistic value head for 1 epoch with learning rate $2 \times 10^{-6}$. After obtaining the attribute-specific uncertain-aware probabilistic value head and base model, we keep them frozen and train the gating layer on Skywork-reward-preference-80k (Liu & Zeng, 2024) for 4000 steps with batchsize 256. During training, we held out 4k data from the dataset as validation set to choose the checkpoint with highest validation accuracy.

RewardBench (Lambert et al., 2024) , our evaluation benchmark for RMs, has 2985 questions and response pairs. For multi-attribute RMs and BT-model RMs, a prediction for a response pair is correct if the RM gives a higher reward to the chosen response than the rejected response. For generative models, RewardBench evaluates them via LLM-as-a-judge (Zheng et al., 2023). If the generative model prioritizes the chosen response than the rejected response, the prediction is seen as correct. To test URM and URME's ability in improving LLMs' generation quality, we evaluate URM and URME with best-of-$n$ sampling (Stiennon et al., 2020) on AlpacaEval (Li et al., 2023).

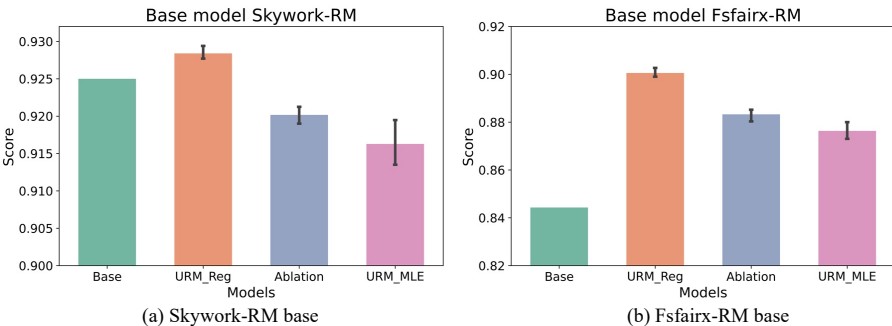

(a) Skywork-RM base        (b) Fsfairx-RM base

Figure 4: RewardBench overall scores for the ablation study. URM_Reg is URM trained with attribute regression while URM_MLE is trained via maximum likelihood estimation. 'Ablation' replaces the uncertainty-aware value head with a linear layer directly predicting the attribute scores.

## 5.2 RESULTS

### 5.2.1 OVERALL RESULTS

Table 1 gives the results on RewardBench. The compared baselines include multi-attribute RMs (Nemotron4-Reward (Adler et al., 2024), ArmoRM (Wang et al., 2024a), SteerLM-RM (Dong et al., 2023b)), BT-model RMs (Skywork-reward (Liu & Zeng, 2024), GRM (Yang et al., 2024), InternLM2-RM (Cai et al., 2024)) and generative RMs (SFR-Judge-r, Gemini-1.5-pro (Google, 2024), GPT-4o (OpenAI, 2024b), GPT-4-turbo (OpenAI, 2024a)). SFR-Judge-r is a chatbot developed by Salesforce based on Llama3.1-70B.

The results on RewardBench confirm URM's strong ability in reward modeling. URM achieves the highest ranking among 8B models and outperforms a number of larger models including Nemotron-4-340B-Reward, also a multi-attribute RM. Except Chat where almost all models have relatively good performance, URM demonstrates improvement over the base model Skywork-8B in all abilities. Especially, compared to ArmoRM which is also a multi-attribute RM with gating layers, URM's better performance shows the efficacy of modeling human preferences as distributions.

### 5.2.2 ABLATION STUDY

Here we study the effect of the uncertain-aware value head and different training methods of URMs. To test the applicability of URM, we initialize URM with two different base models: Skywork-RM (Liu & Zeng, 2024) and Fsfairx-RM (Dong et al., 2023a). Fig. 4 gives the results of our ablation study. 'Ablation' refers to the model with a value head to directly map hidden states to score values instead of sampling in URM. All other components of Ablation are kept the same as URM. URM_Reg is an URM trained with the attribute regression loss function in Eq. 5, while URM_MLE is trained via maximum likelihood estimation. Since the dataset Helpsteer 2 for our attribute prediction has already been used in the base model Skywork-RM, Ablation and URM_MLE do not demonstrate improvement over the base model, and only URM_Reg surpasses the base model by modeling the preference distributions. But with base model Fsfairx-RM not trained with Helpsteer 2 previously, all our models demonstrate significant improvement over the base model. Especially, URM trained via attribute regression significantly outperform its counterpart with MLE loss. However, although URM_Reg has better performance in prioritizing chosen responses over the rejected, URM_MLE demonstrates better uncertainty quantification and distribution modeling ability. We theoretically illustrate this phenomenon in the appendix B. Thus, for other studies involving uncertainty quantification, we use URMs trained via the MLE loss.

Fig. 4 indicates regression-based training methods achieve higher scores on RewardBench. This could potentially be credited to the high quality of Helpsteer 2 dataset, which is meticulously processed and derived from Helpsteer (Wang et al., 2023). This quality enables even the simplest direct attribute regression to deliver substantial performance improvements, as shown by the Ablation with base model Fsfairx-RM. However, the introduction of noise via the sampling-based scores in URM_Reg makes URMs more robust in distinguishing between chosen and rejected responses. Despite this, we anticipate that URM_MLE would prove more successful on real-world datasets,

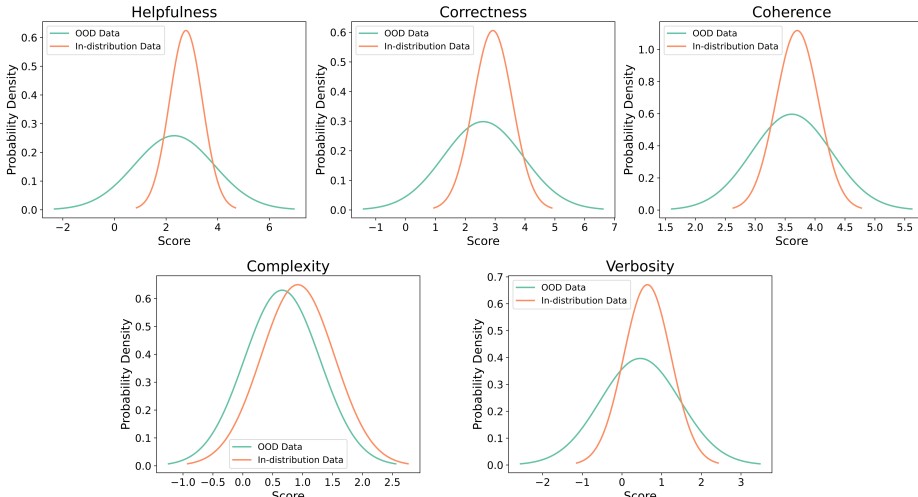

Figure 5: Attribute score distributions modeled by URM. Means and variances are estimated and averaged by OOD and in-distributional samples separately.

which often encompass a wide spectrum of data quality, so that modeling distributions of the scores becomes more necessary.

### 5.2.3 UNCERTAINTY QUANTIFICATION

Now we study the uncertainty quantification of URM and URME and how they behave when dealing in-distribution and OOD data. Given the challenge inherent in identifying what precisely is OOD for LLMs, we adopt numeric calculations as simulated OOD data. This is because LLMs are known to underperform in this skill area. Details are given in the appendix A.2.

Fig. 5 gives the attribution score attributions of OOD and in-distribution data modeled by URM. Due to the lack of knowledge to accurately evaluate the OOD data, the modeled distributions for OOD data have significantly larger variance and are much closer to uniformity than for in-distributional data. Therefore, this uncertainty quantified by the variance can serve as an informative tool for identifying and filtering out OOD data, where reward models exhibit a tendency towards making uniform guess than providing an accurate evaluation. This strategy ensures the evaluated outcomes are both more dependable and robust.

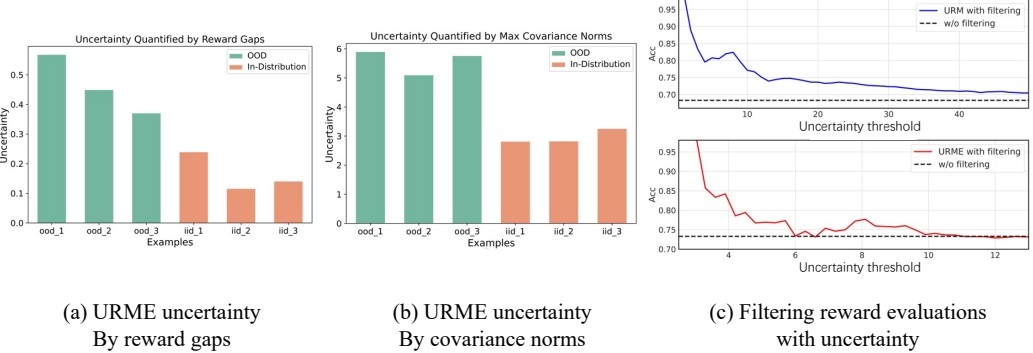

(a) URME uncertainty
By reward gaps

(b) URME uncertainty
By covariance norms

(c) Filtering reward evaluations
with uncertainty

Figure 6: URME's uncertainty quantification by (a) maximum reward gaps and (b) maximum covariance norms. Larger discrepancies exist among the URM ensemble when dealing with OOD data. (c) Reward evaluation accuracy when uncertainty of prompt-response pairs is within the threshold. 2000 test questions are used in this evaluation. The results confirm that uncertainty of URM and URME is able to indicate reliability of reward predictions.

We quantify uncertainty in URME with two metrics: maximum reward gaps in Eq. 7 and maximum covariance norms in Eq. 8. URME uncertainty quantification results are given in Fig. 6(a), (b).

Quantified uncertainty under two metrics both indicate that URME is substantially more uncertain on OOD data. The results confirm that when the URMs lack relevant knowledge to make accurate reward predictions, they will diverge with each other, demonstrating significant discrepancies.

To test whether quantifying uncertainty is able to improve reliability of the evaluated rewards, we utilize 2k prompts from our held-out validation set (described earlier) and evaluate their rewards and uncertainties. Fig. 6(c) gives URM and URME's evaluation accuracy with uncertainty-based filtering. In this setup, prompts and responses (either the chosen or rejected) with uncertainty larger than the threshold are filtered out. URME use reward gaps to quantify uncertainty and URM's uncertainty is quantified by summation of each attribute distribution's variance. The results validate our claim that reward predictions with low uncertainty are more reliable than those with high uncertainty. Therefore, through uncertainty quantification, we can decide whether the reward predictions are unreliable and need to be filtered out, which will lead to improved reliability of reward evaluations.

### 5.2.4 GENERATION RESULTS IMPROVEMENT

We evaluate URM and URME's ability in improving LLMs' generations with best-of-$n$ sampling on AlpacaEval (Li et al., 2023). Specifically, we prompt Llama3-8b-Instruct model with 805 questions from AlpacaEval for $n$ times, evaluate each response's reward and choose the response with highest reward (highest average reward for URME) as the answer. The answer is then compared against the reference answer provided by the benchmark with LLM-as-a-judge (Zheng et al., 2023). In LLM-as-a-judge, we use the official prompt of AlpacaEval and GPT-4-0125-preview as the judge. Details are given in the appendix A.3.

Table 2: Win rates of Llama3-8b-Instruct against reference answers on AlpacaEval.

| Evaluator | Best-of-1 | Best-of-4 | Best-of-8 | Best-of-16 | Best-of-32 | Best-of-64 |
|-----------|-----------|-----------|-----------|------------|------------|------------|
| URM | 81.2% | 82.7% | 83.1% | 83.5% | 83.9% | 85.3% |
| URME | 81.2% | 83.4% | 84.5% | 85.5% | 86.6% | 86.4% |

Table 2 gives the results of using URM and URME to improve generation quality. In our experiments, the baseline model Llama3-8b-Instruct achieves $81.2\%$ win rate (best-of-1). As the number of samples increases, both URM and URME are able to evaluate the quality of responses, thus ameliorating the baseline model's generative performance. Furthermore, URME is able to consistently outperform URM in improving generation quality, as it combines the strength of several independent models and mitigates biases during reward evaluation (Coste et al., 2023; Eisenstein et al., 2023).

## 6 CONCLUSION AND FUTURE WORK

In this paper, we study the uncertainty issue in reward modeling for LLMs. Uncertainty-aware Reward Model (URM) and Uncertain-aware Reward Model Ensemble (URME) are proposed to model and quantify the uncertainty during reward modeling. Unlike previous methods that deterministically map hidden states to reward scalars, URM and URME model the distribution of rewards and evaluate prediction confidence by uncertainty quantification. Notably, among RMs with 8B or smaller model size, URM achieves state-of-the-art performance on RewardBench, surpassing a number of larger models. Empirical evidence further validates that through uncertainty quantification, URM and URME can effectively evaluate their level of knowledge for input data, leading to more reliable reward predictions.

The limitation of our paper is that our experiment for generation improvements is limited to best-of-$n$ due to the limit of computation resources. In the future, we plan to introduce URM and URME to prevailing LLM alignment methods like RLHF Ouyang et al. (2022) and iterative DPO Yuan et al. (2024). Another direction to look at is model merging for URMs in the weight space (Ramé et al., 2024), which demonstrate competitive efficiency and robustness compared to ensembles. A simple empirical study to RM merging with URMs is included in the appendix C.1.

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
