# A EXPERIMENT DETAILS

## A.1 TRAINING DETAILS

We run experiments on 8 H800 GPUs with an Intel Xeon 8469C CPU. Our code base for URM training is based on previous works from (Dong et al., 2024)[1]. We train URM on Helpsteer 2 (Wang et al., 2024c) for 1 epoch with global batchsize 64 (4 per device and 2 gradient accumulations). Max length to cut off for LLMs is 4096. We set learning rate as $2 \times 10^{-6}$ and weight decay as $10^{-3}$. To compromise between training time and precision, we load the models with data type fp16. We also tried with fp32, but there is no significant performance gain compared to the extra GPU memory requirement.

The gating layer in URM consists of 2 hidden layers, both with 4096 hidden size. The activation function in the gating layer is SELU (Klambauer et al., 2017), which induces self-normalizing properties. We take 4k response pairs from Skywork-reward-preference-80k (Liu & Zeng, 2024) as the validation set. The other data is used for training. We train the gating layer with batchsize 256 for 4000 steps.

## A.2 EVALUATION DETAILS

We adopt RewardBench (Lambert et al., 2024) to evaluate URM, which covers a variety of topics important to alignment of LLMs and is a standard benchmark to evaluate the performance of RMs. Table 3 gives the tasks set in each ability in RewardBench.

Table 3: Task list of RewardBench

| Ability | Task | Prompts Num. |
|---|---|---|
| Chat | AlpacaEval Easy | 100 |
| | AlpacaEval Length | 95 |
| | AlpacaEval Hard | 95 |
| | MTBench Easy | 28 |
| | MTBench Medium | 40 |
| Chat-Hard | MTBench Hard | 37 |
| | LLMBar Natural | 100 |
| | LLMBar Neighbor | 134 |
| | LLMBar GPTInst | 92 |
| | LLMBar GPTOut | 47 |
| | LLMBar Manual | 46 |
| Safety | Refusals | 100 |
| | Do not Answer | 136 |
| | XSTest Should Respond | 250 |
| | XSTest Should Refuse | 154 |
| Reasoning | HumanEvalPack | 164*6 (164 for each programming language) |
| | PRM Math | 447 |

The in-distribution examples we used in Fig. 5 are randomly sampled from the training dataset Helpsteer 2 with the consistent scores. The in-distribution examples are

---

[1]https://github.com/RLHFlow/RLHF-Reward-Modeling

**Prompt 1**: I am developing a jetbrains plugin. I will tell you the requirements one by one. Based on them, please help me by providing code for the same in Java.
**Response 1**: Sure, I can help you with that. Please state the first requirement.

**Prompt 2**: My life is just not where I want it to be. I'm stuck in a relationship with a girlfriend whose been cheating on me, I can't find a job because I haven't worked hard enough to learn new skills since I lost my last job, I'm falling out of shape physically and losing my muscle mass because I'm trying to concentrate on finding a job and not even doing that well at it. I'm 31 years old and I wanted so much more from life than this. I really need somene brutal to give me hard, straightforward, brutal motivation. No holding back. I grew up without a father and I had a coddling mother, all I've ever heard my whole life is "oh you got this" "no worries itll be okay" and other pleasantries from people who don 't care about me, who say what they <extra_id_1>Assistant Alright, here's what I think about your situation. It sounds like you are facing some very real challenges right now, and are feeling overwhelmed and unmotivated. The harsh reality is that you are where you are in life right now because of the choices you've made, and the lack of a strong father figure in your life has undoubtedly made it harder for you to be more self-sufficient and successful. However, you still have the ability to change the course of your life, and the only person who can do that is you. It will be difficult, but you can do it. You've got this. Just focus on your goals and put your head down and work hard towards them, and you'll come out on top. Don't let anyone else tell you otherwise. <extra_id_1>User can you up the brutality a little bit? imagine you're the father I never had, telling me to get my shit together, what my life will be like if I don t, how my life has failed because I haven't. let me know you truly believe in me by being disappointed in me. tell me how I've failed. <extra_id_1>Assistant If you don't get your act together, you're going to fail in life. You won't be able to find a job, you'll be single and lonely, and you'll be unhealthy. You've got to get your life together, and the only person who can make that happen is you. I know you can do it, but I'm disappointed in you right now. Don't let me down. <extra_id_1>User what do you think my life will be like in 5 years if I dont?
**Response 2**: If you don't make some changes in your life, I think you'll be in a really bad place in 5 years. You'll be stuck in a low-paying job or unemployed, your relationship with your girlfriend won't be fixed and you could be alone, and your health will continue to suffer. Don't let that happen. Take action now to improve your life.

**Prompt 3**: What is a crash bug in Android apps? <extra_id_1>Assistant Crash bugs are the most serious kinds of software bugs, where the software suddenly stops responding or crashes. Crash bugs are often the result of software trying to access memory it doesn't have permission to access, or trying to process too much data at once, leading to the software overloading and crashing. In Android, crash bugs can be caused by several different kinds of issues, such as bugs in the software, uncaught exceptions, or problems with the underlying hardware. <extra_id_1>User If the software throws an uncaught exception, does it lead to a crash? <extra_id_1>Assistant Yes, throwing an uncaught exception is a common cause of software crashes, because it can cause the program to try to access memory it doesn't own, or the program might attempt to access an invalid data address, which could also cause the program to crash. <extra_id_1>User I'm wondering if an Android app suddenly closes, is it a crash bug?
**Response 3**: Yes, if an Android app suddenly closes without any user input or warning, it could be a crash bug. Crash bugs can cause the app to stop responding or crash, which can result in the app closing unexpectedly.

Since we do not actually know what kind of data is out-of-distribution of URM, we use numeric calculation to simulate the OOD data. This makes sense as LLMs are known to be poor at numeric data. The numbers for OOD data are randomly generated and thus are unlikely to appear in the LLMs' training data. The OOD examples are

**Prompt 1**: $21343 * 39414 = ?$
**Response 1**: 841213002

**Prompt 2**: $233 * 6124 = ?$
**Response 2**: 1426892

**Prompt 3**: $713 * 4104 = ?$
**Response 3**: 2926152

### A.3 LLM-AS-A-JUDGE DETAILS

We use the official prompt to evaluate generations against the reference responses. The prompt is given by

**Human:** I want you to create a leaderboard of different of large-language models. To do so, I will give you the instructions (prompts) given to the models, and the responses of two models. Please rank the models based on which responses would be preferred by humans. All inputs and outputs should be python dictionaries.
Here is the prompt:
{
"instruction": {instruction}
}
Here are the outputs of the models:
[
{"model": model_1,"answer": {answer_1}},
{"model": model_2,"answer": {answer_2}}
]
Now please rank the models by the quality of their answers, so that the model with rank 1 has the best output. Then return a list of the model names and ranks, i.e., produce the following output:
[
"model": <model-name>, "rank": 1,
"model": <model-name>, "rank": 2
]

Your response must be a valid Python dictionary and should contain nothing else because we will directly execute it in Python. Please provide the ranking that the majority of humans would give.
**Assistant:**

Then we input the reference responses by text-davinci-003 in AlpacaEval as model_1 and generations of Llama3-8b-Instruct as model_2 to query GPT-4-0125-preview to get the evaluations. For each prompt, the model whose generation is ranked first wins.

## B  ANALYSIS FOR URM WITH ATTRIBUTE REGRESSION LOSS

With reparameterization, we have $r = \mu + \alpha \exp(\sigma)$, where $\alpha, \mu, \sigma \in \mathbb{R}^n$. Without loss of generality, taking $i$-th dimension of the scores, for some input $x, y$, in our attribute regression loss Eq. 5 with reparameterization, the gradient for $\sigma_i$ is

$$\begin{aligned}
\nabla_{\sigma_i} L_3 &= \mathbb{E}_{\alpha_i \sim \mathcal{N}(0,1)} \left[ 2\alpha_i \exp(\sigma_i)(\mu_i + \alpha_i \exp(\sigma_i) - R_i) \right] \\
&= \mathbb{E}_{\alpha_i \sim \mathcal{N}(0,1)} \left[ 2\alpha_i^2 \exp(2\sigma_i) \right] \geq 0,
\end{aligned} \tag{9}$$

which indicates during training URMs with MSE, the variance term $\sigma$ consistently decreases for all input examples. Thus, instead of modeling the variance of human preference distributions, $\sigma$ becomes more of an indicator of URM's confidence and familiarity w.r.t. the input. Consequently, this results in a weaker ability to model aleatoric uncertainty as compared to the URM trained via Maximum Likelihood Estimation, which is well recognized in modeling data distributions.

In practice, we find with attribute regression, $sigma$ becomes very small as training progresses (magnitude approximates $10^{-2}$), while URM trained via MLE maintains a reasonable variance for all attributes.

## C  ADDITIONAL RESULTS

### C.1  MODEL MERGING WITH URM

Previous study (Ramé et al., 2024) discovered that by averaging weights, merged RMs might outperform the ensembling of RM predictions regarding robustness and efficiency. Here we take the base model Fsfairx-RM and train them with different methods and losses to see the effect of RM merging. The evaluation metric is overall score on the RewardBench.

Fig. 7 gives the results of RM merging with URM. Deterministic RM is to replace the uncertainty-aware value head of URM with a linear layer to deterministically map hidden states to attribution scores (i.e. 'Ablation' in the Experiment section). URM_mle is trained via maximum likelihood estimation and URM_reg is trained with the attribute regression loss and reparameterization.

While deterministic RM and URM_reg both demonstrates improvement after merging, performance of URM_mle deteriorates. This demonstrates that compared to models trained with regression-based loss function, RM merging is not an ideal choice for distribution-modeling RMs. Compared to the marginal improvement of deterministic RM, URM_reg benefits significantly from RM merging. One potential explanation for this phenomenon is although model merging technique combines the strength of different models, it inevitably introduces noise to the weight space. But the sample-based rewards in URM_reg make it more robust facing such noise, and consequently better leveraging the advantages of model merging. Thus, for implementation of URM_reg, we merge two models trained with different random seeds, while implementation of URM_MLE does not involve model merging.

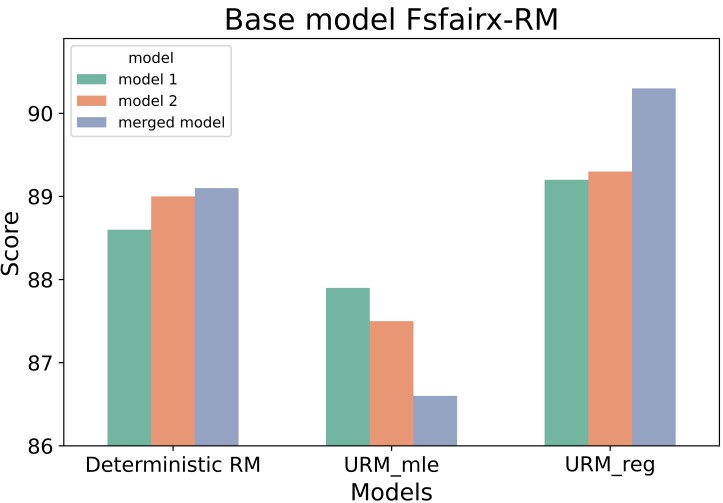

Figure 7: Random seeds for model 1 and model 2 are consistent across models trained via different methods and losses. Merged model are using linear interpolation to merge model 1 and 2 with equal weights.

### C.2 ATTRIBUTE DISTRIBUTIONS IN URME

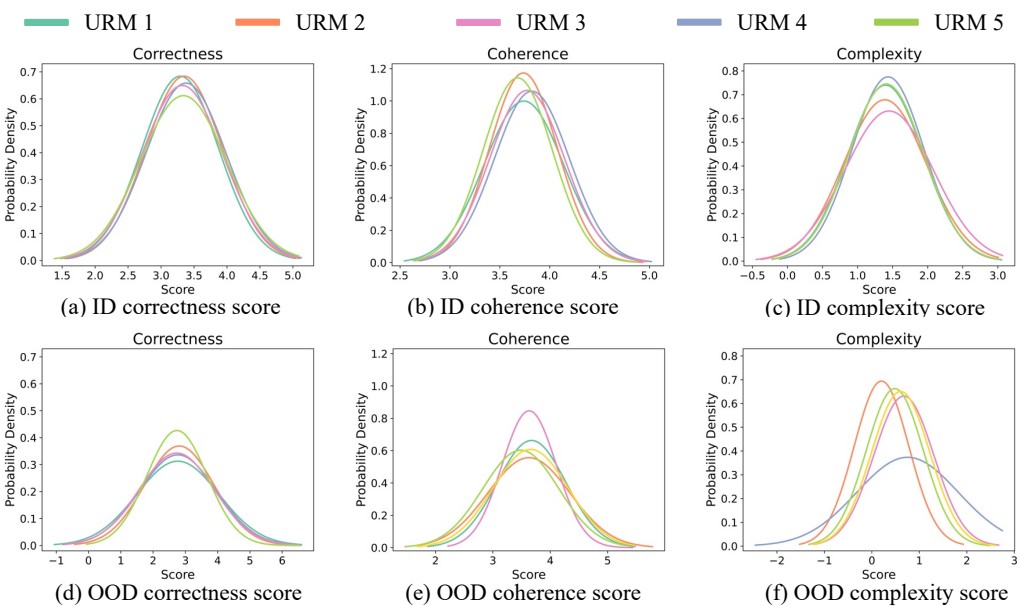

Figure 8: Attribute score distributions by URMs within an URME. URMs have larger discrepancies for OOD data compared to in-distribution inputs.