# OpenReview forum: "Uncertainty-aware Reward Model: Teaching Reward Models to Know What is Unknown"
_ICLR.cc/2025/Conference — ICLR 2025 Conference Withdrawn Submission_

### Official Review · Reviewer_qSrc · 2024-10-31

**Soundness:** 2
**Presentation:** 3
**Contribution:** 2
**Rating:** 3
**Confidence:** 5

**Summary:**

The paper tackles the problem of estimating the reward model uncertainty from both labelling noise and the information missing in the dataset. Uncertainty due to labelling noise is modeling at the individual reward model level and uncertainty due to missing information with an ensemble of reward models. The authors identify that reward model aleatoric uncertainty cannot be estimated when training with the Bradley-Terry objective. Instead the authors turn to multi-attribute reward model learning as a training objective for which uncertainty can be estimated. Based on the RewardBench leaderboard, the authors find that the uncertain aware reward models out perform the reward models that are not. The authors additionally demonstrate that the uncertainty quantification can be used to detect OOD data and can be used LLM generation in a best-of-n setting on AlpacaEval

**Strengths:**

* The work takes up the important challenge of incorporating reward model uncertainty into policy learning. Previously reward model uncertainty has been used to guide policy exploration, but this is a different and important avenue to explore.
* The paper is well written and the need for uncertain aware reward models is well motivated.

**Weaknesses:**

** High level **
The main issue with the paper right now is their training data for the gating layer has leakage from the test benchmark and a main point of their paper is URM/URME being SoTA. Other issues stem from sharing an appropriate background to help the reader understand the key motivations and take aways. Some of the results are not as meaningful as they could be in the absence of more extensive experiments to understand the impact of URM/URME on downstream tasks especially compared to non uncertain aware reward models.

** The Details **

* The skywork-reward-preference-80k (used to train the gating layer) has been found to have overlap with RewardBench, which is the evaluation benchmark used in this paper. See the note [here](https://huggingface.co/datasets/Skywork/Skywork-Reward-Preference-80K-v0.1). The updated dataset is v0.2 [here](https://huggingface.co/datasets/Skywork/Skywork-Reward-Preference-80K-v0.2). The new dataset was added to HuggingFace at the start of Oct. 2024. As large parts of the paper claims relies on comparisons to leader board results, the results need to be rerun with the updated and cleaned version of the dataset. While this is not a fault of the authors, the results cannot strongly support the claims when their training data overlaps with the benchmark data.
* It is not clear why uncertainty cannot be estimated when training with the Bradley-Terry object.help
* The work relies heavily on knowledge of multi-attribute reward model methods and uncertainty quantification methods. However, background information is not provided on these areas to help the reader understand what the proposed method is doing and the motivation for that. The transition from the Bradley-Terry approach to the multiple attribute approach is particularly jarring as the data is set up in completely different ways and the prediction targets are very different. More of the key differences should be discussed, such as the multi-attribute data is frequently labelled with a scalar score per attribute. For uncertainty quantification, it would be helpful to have more details about why the aleatoric and epistemic approaches are reasonable earlier instead of more than halfway through the paper. Also to give a description of what the reparameterization technique is.
     * When talking about Equation 4 and the label, it is important to state what the label is a where it comes from. Also R in Equation 4 needs to be defined.
* The justification for why reward models cannot quantify uncertainty needs to be made for clear. It seems like some steps to reach the final conclusion are skipped. If the take away is the loss term is driven by the difference in $r_w$ and $r_l$ making it so the exact value of $\sigma_w$ does not matter to drive the loss to zero, this should be more explicitly said. Rephrasing "...RMs still cannot quantify the uncertainty even equipped with a probabilistic value head" could be modified to include a statement along the lines of, "...$\sigm_w$ cannot be accurately learned due to the difference between $r_w$ and $r_l$ being the primary drives of the loss approaching 0, therefore the RMs cannot quantify the uncertainty even equipped with a probabilistic value head."
* Only one benchmark is used to evaluate (RewardBench). A quantification of the alignment tax associated with these methods are missing. This can be assess through comparing performance of the base versus aligned models on NLP tasks such as MMLU.
* At the end of Section 5.2.3 the authors state, in reference to filtering data based on it being OOD for a reward model, "...which will lead to improved reliability of reward evaluations." However, there are not quantitative results to back this claim up.
* The filtering results are not particularly meaningful as there is no measure of how the proposed filtering method impacts downstream performance.
* For the "Generation Results Improvement" the Table 2, the model used to produce the reference answers is not specified. Additionally, without a baseline that does not have uncertain aware reward models it is not possible to tell how the uncertain aware reward models impact generation performance/quality.
* For the "Generation Results Improvement" there should be evaluations on some standard NLP benchmarks like MMLU to quantify the alignment tax with URM/URME versus reward models that are not uncertain aware.
* typographical issues (not impacting score):
    * everywhere "Ablation" is referenced for the Section 5.2.2 Ablation Study results, the first first quotation is backwards. Make sure to use "`".
    * on line 490, a validation set is specified as "described earlier". Please add a pointer to the section.
    * In table 1, please label the models as multi-attribute, BT, or generative. Also please bold the best result in each column.
     * line 280, "parameter$\alpha$" is missing a space
     * Figure 2, please use a legend for the blue and orange lines instead of arrows pointing to labels.
     * line 268, "Thourgh" -> "Through"

**Questions:**

* To what extent is the stochasticity of preferences due to actual stochasticity versus partial observability? If preferences are sensitive to mood then knowing this the preference process wouldn't be stochastic.
* Please elaborate on your offline versus online RL distinction. It sounds like you are saying it is offline RL because the reward learning data is collected separate from policy learning. But that doesn't mean the data used to train the policy is collected separate from policy training.

---

### Official Review · Reviewer_fvAc · 2024-11-01

**Soundness:** 2
**Presentation:** 2
**Contribution:** 3
**Rating:** 5
**Confidence:** 4

**Summary:**

The authors propose an ensemble method, denoted URME, to uncertainty-aware reward modeling, showing good results on RewardBench. Ensemble members feature an uncertainty layer that outputs the mean and standard deviation for each reward attribute, and also learns attribute weights in a separate step.

**Strengths:**

Nice results on RewardBench.

Figures are well done and help understand the different contributions and experimental results from the paper.

Fig 5 showing higher variance for OOD than ID, and similar means is quite encouraging for uncertainty quantification.

**Weaknesses:**

A key limitation from my perspective is that the design choices and details of URM / URME could be motivated much better:
* Why sampling once only from each URM? Also why not using the full distribution to derive a notion of uncertainty? If I follow the aleatoric vs epistemic argument, then epistemic uncertainty could be measured using a quantification of the overlap between the different probability densities (taking into account variances to avoid encouraging large variance). The current choice should thus be explicited better.
* Isn’t equation 7 problematic? I.e. one URM could wrongly attribute a very high (or very low) reward whereas all other URMs rightfully estimate the reward.
* Same for equation 8 with a URM outputting a large variance..
Incidentally, I have doubts regarding the chosen method for mapping ensembles of probability distribution parameters to uncertainties.

Some key experiments and ablations are missing:
* learned reward weights (i.e. gating) versus uniform, expert (i.e. same as existing work for instance) or random weights
* clearly disentangle the benefit of ensembling from that of uncertainty quantification
    * only done for BoN, not for uncertainty quantification nor accuracy
    * also evaluating individual URMs from URME would be a quite interesting addition
* study the correlation between high uncertainty from URM / URME and loss / evaluation of answers
* The results obtained in Table 2 are good, but I would like to see URM / URME applied in a case where Bo1 win rates are more even. Showing results for higher values of N (and subsequently, whether this saturates or not) would be a compelling addition
* How many URMs in URME? Do we have some evaluation of performance depending on the number of ensembled reward models?
* Though RLHF / direct alignment from preferences can arguably fall outside of the paper scope, conducting such an experiment would make the paper more compelling

When URM vs URME is not clear in the paper, especially in experiments.

On math notations and statements:
* $sigmoid$ -> $\sigma$ is a popular choice that would make expressions cleaner
* “We denote the reparameterization parameter $\alpha \sim \mathcal{N}(0, 1)$, and thus the sampled reward $r = \mu + \alpha
 \exp(\sigma)$”
    * the fact that this is coherent with the statement line 239 should be quickly explained

Clarity can be improved:
* The introduction does not discuss RLHF at all!
* “LLM alignment typically consists of three stages (Ouyang et al., 2022): supervised fine-tuning (SFT), reward modeling and proximal policy optimization (PPO) (Schulman et al., 2017).”
    * this is misleading! PPO is only one instance of RLHF methods. DPO / REINFORCE and many other relevant works should be mentioned.
    * this statement is reductory and should be adapted.
* /!\ The description of PPO is insufficient as of now! The clipped loss / GAE and other crucial details of PPO are currently being left out.
* “To achieve alignment, LLMs rely on feedbacks from reward models (RM), where the feedbacks are provided in the form of rewards (Singhal et al., 2023a; Cui et al., 2024; Kasneci et al., 2023).” -> authors could mention RL(HF) as the reason for this, to best set the context
* “ Moreover, there is no other information to validate the reliability of these reward predictions.” -> please clarify
* “Therefore, introducing uncertainty to reward modeling improves modeling capacity of RMs and enhance reliability of the reward predictions” -> this statement either needs to be backed up by some data (and proper citing) or adjusted
* “URM is equipped with an uncertainty-aware value head to model the distributions of multiple attributes within human preferences” -> are uncertainty-aware value heads known in the community (then needs references) or a contribution of this work? this needs clarifying
* “Contributions of this paper include:” -> are there other main contributions? “Our main contributions are” would be a clearer formulation.
* “Reparameterization technique is adopted to enable gradient back-propagation” this should be explained in more details
* “Gating” is in my opinion a poor choice of name to describe the learning of coefficients for the different rewards
* Not clear to me whether learning the reward weights is a contribution or not, I would be surprised that this was not explored in the literature

Writing is clearly sub-par in the current state. Some examples:
* “In this paradigm, RMs fundamentally decides the efficacy of alignment, as they primarily steer the LLMs through feedback”
* “We demonstrate that with the popular bradley-terry-model loss function” -> Bradley-Terry loss
* “During reward evaluation, filtering strategy can be applied to prompt-response pairs with high uncertainty”
* “Furthermore, results of best-of-n sampling validates that URM and URME can effectively enhance the generation quality of LLMs” -> performance instead of result?? also need to clarify that best-of-n sampling is applied on top of a given policy
* “log P(Ri|x, y) is the log-probility of Ri from the parameterized distribution N (µ_i, exp(2i)). Thourgh MLE”
* “traditional RMs can only provide a fixed reward estimation and fails to represent the real preference”
* “However, we show that introducing the probabilistic value head and the sample-based reward to RMs with BT-loss, the aleatoric uncertainty still cannot be quantified.”

**Questions:**

See other questions above.

Why use SELU? (l. 362)

How are examples selected for Fig 6 a) b)? This could be misleading if examples are not representative, so I would advocate to plot the uncertainty distribution instead.

In Fig 6 c) it would be interesting to plot the number of examples that are not filtered out

Could it be possible to learn reward weights jointly with the individual URMs? This would make the method more compelling as it would not require an extra, separate phase of training.

---

### Official Review · Reviewer_JUf1 · 2024-11-01

**Soundness:** 2
**Presentation:** 3
**Contribution:** 2
**Rating:** 3
**Confidence:** 4

**Summary:**

Major contributions of the paper are as follows:
- The paper identifies that Bradley Taylor loss function (which uses preferences over pairs of rollouts) is not suitable for training uncertainty aware reward models.
- The paper therefore proposes using multi-attribute RMs (for which training data is in form of ratings) for which straightforward training objectives exist for teaching RMs to express uncertainty.
- The paper goes on to empirically show that uncertainty RMs and their ensembles generally do better than their deterministic counterparts.

**Strengths:**

- I liked the results and experiments in section 5.2.3
- Paper was relatively well-written and easy to follow
- The paper is honest about its limitations, and generally the paper does not 'oversell'. I liked this about the paper.

**Weaknesses:**

- Very limited novelty; the idea of using uncertainty over RMs has already been explored in the prior literature.
- The paper is also not really proposing any novel way of parameterizing uncertainty imo.
- The improvement over baselines (table 1) are relatively minor.
- Table 2 has no baselines. Baslines from table 1 can be reused here imo.
- The paper is only using BoN; RL optimization exerts much stronger optimization pressure so experiments done there could potentially have been more interesting if the method proposed was shown to clearly mitigate overoptimization (this is the **major** limitation in my opinon, BoN resluts are just not very informative here unfortunately imo).

**Questions:**

1. Why jsut aleatoric uncertainty and not aleatoric + epistemic uncertainty? This is a key claim in the paper but it is not substantiated in anyway and on the surface feels false to me.

> Considering RMs act similarly to the reward part of the dynamics model in MBRL, aleatoric uncertainty within human preferences can be captured by outputting the parameters of a parameterized distribution.

2. The paper shows that the BT loss is incapble of teaching RMs to quantify uncertainty when output distribution is parameterized as normal. How fundamental is this issue to BT loss? My sense is that for symmetric distributions (like normal, or laplace) its true, but probably not if you allow the output distribution to be asymmetric (like Gumbel?). Authors should add more discussion on this.

3. The authros use gating layer to combine mean values of attribute reward models; how is the uncertainty over individual attribute RMs combined into a scalar value? If its not combined, how its used? Equations 7 and 8 are supposed to be about this but I am not sure I understand what r_i and r_j in those equations are. Is the 'ensemble' being used to mean an ensemble of 'individual attribute RMs' or are there mutliple RMs being trained for individual attributes that are then ensembled over?

4. Not a question, but please put appendix and paper in the same pdf as is the tradition for ICLR papers. Moving between pdfs is very annoying.

---

### Note · Authors · 2024-11-23

I have read and agree with the venue's withdrawal policy on behalf of myself and my co-authors.